# Risk Factors for Locomotive Crew Members Depending on Their Place of Work

**DOI:** 10.3390/ijerph19127415

**Published:** 2022-06-16

**Authors:** Elena A. Zhidkova, Ekaterina M. Gutor, Inga A. Popova, Victoria A. Zaborova, Kira Kryuchkova, Konstantin G. Gurevich, Natella I. Krikheli, Katie M. Heinrich

**Affiliations:** 1UNESCO Chair, A.I. Yevdokimov Moscow State University of Medicine and Dentistry, 127473 Moscow, Russia; genmedc@gmail.com; 2Central Directorate of Healthcare—The Branch of Joint Stock Company “Russian Railways” (JSC “RZD”), 123557 Moscow, Russia; gutorem23@mail.ru; 3Institute of Biodesign and Complex Systems Modeling, I.M. Sechenov First Moscow State Medical University (Sechenov University), 119991 Moscow, Russia; ainessa77@gmail.com; 4Institute of Clinical Medicine, I.M. Sechenov First Moscow State Medical University (Sechenov University), 119991 Moscow, Russia; zaborova.va@mipt.ru (V.A.Z.); kira.kruchkova@mail.ru (K.K.); 5Sports Adaptology Lab, Moscow Institute of Physics and Technology (National Research University), 141700 Dolgoprudniy, Russia; 6Department of Public Health, Research Institute of Healthcare Organization and Medical Management of Moscow Department of Healthcare, 115184 Moscow, Russia; 7Department of Clinical Dentistry, A.I. Yevdokimov Moscow State University of Medicine and Dentistry, 127473 Moscow, Russia; nataly0088@mail.ru; 8Department of Kinesiology, Kansas State University, Manhattan, KS 66506, USA; kmhphd@ksu.edu

**Keywords:** train driver, assistant driver, risk factor, railway

## Abstract

Purpose: The purpose of this study is to examine the prevalence of workplace exposure, behavior, and individual health conditions, along with resulting medical activity among locomotive crew members depending on their place of work. Patients and methods: Participants included 5585 train drivers and 3723 train drivers’ assistants (7% of the total train operators in the Russian Federation). Measured height and weight were used to calculate body mass index (BMI), and waist circumference, pulse rate, and blood pressure were also measured. The risk assessment was conducted using the STEPS tool. The level of commitment to a Healthy Lifestyle was assessed based on World Health Organization recommendations. Occupational risk factors were surveyed. Morbidity was analyzed over the past year. Results: The lowest frequency of work exposure risk factors was found for employees of the Trans-Baikal railway; the highest was among Krasnoyarsk, North, and South-East. The participants from the Far East and October Railways had the lowest self-reported frequency of behavioral risk factors. The participants from the Eastern Siberian, October, and Southern Urals railways had the lowest occurrence of individual health conditions. The participants from the East Siberian, Far East, Kuibyshev, and Sverdlovsk railways were the least likely to visit their doctor, take temporary disability leave, or be hospitalized. The total assessed Healthy Lifestyle commitment index was higher for participants from the Far Eastern and Southern Urals railways. The participants from the Moscow and October railways were the least committed to a Healthy Lifestyle. Conclusions: Significant differences exist between risk factors and Healthy Lifestyle commitment between railways. Future research should examine changes due to a new corporate health program introduced in 2020.

## 1. Introduction

Lifestyle factors determine approximately 50% of a person’s likelihood of developing chronic non-communicable diseases, which are the main cause of temporary disability, disability, and adult mortality [1,2]. Most of the risk factors for chronic non-communicable diseases are preventable through prevention programs that influence habitual behavioral patterns [3,4].

According to the literature, the simultaneous impact of several risk factors for chronic non-communicable diseases is particularly detrimental to health [5]. On the other hand, the absence of such risk factors can be seen as a commitment to a healthy lifestyle (HLS) [6,7]. Increased commitment to an HLS is also a measure of the effectiveness of preventive interventions [8].

The most prevalent chronic non-communicable diseases are cardiovascular diseases [9]. Locomotive crew members have increased cardiovascular risk due to occupational (workplace) exposures, psycho-emotional factors, and increased acoustic load [10,11]. At the time of this study, only men could manage locomotives in the Russian Federation (RF), and RF men had greater cardiovascular disease risk than RF women [12].

Russian Railways is one of the biggest railway companies in the world and covers the majority of the country. Historically there were 16 independent railways, but during the last few years, the Kaliningradskay railway became part of the Moscow one (https://company.rzd.ru/ru/9349/page/105553?type_id=3; accessed on 1 April 2022). Railways are different due to climate, the distance between stations, traffic, and other conditions [13]. There are several international publications about the effects of various tasks, such as military task performance in soldiers [14], national differences in work stressors [15], and so on [16,17]. However, we did not find such research available for railway workers.

Thus, the purpose of this study is to examine the prevalence of workplace exposure, behavior, and individual health conditions, along with resulting medical activity among locomotive crew members from different railways.

## 2. Material and Methods

This study was approved by the inter-university ethics committee (Approval #07-19 from 18 July 2019). This paper reports on the part of a series of studies on the health aspects of railway safety in the RF between 2017 and 2019. A detailed description of the research methodology has been previously published in Russian by the authors [18,19]. The tables and results presented in this paper have not been previously published.

The study materials included questionnaires for 5585 train drivers and 3723 drivers’ assistants who gave informed consent (7% of the total train operators in the RF). All of the participants consented to have their data obtained, including that from their personal medical files, as part of the consent process.

A quota sample was used to select respondents. Quotas were carried out according to the following criteria: territorial affiliation, functional branches, and age. The data underwent the procedure of “weighting” based on the number of employees of each railway, functional branches, and different age categories.

Due to the Russian legacy, only men worked as train drivers or drivers’ assistants before 1 January 2022. Thus, all participants included in the study were men. The mean age was 35.8 ± 9.3 years.

Anthropometric measures were conducted in person. The body mass index (BMI) of the participants was calculated from the measured height and weight, and waist circumference, pulse rate, and blood pressure were also measured. Participant values of total cholesterol and high blood glucose were provided from their medical documentation. Behavioral risk factors were assessed using the STEPwise Approach to Non-Communicable Diseases Risk Factor Surveillance by the World Health Organization (WHO) [20]. This is a simple, standardized method for collecting, analyzing, and disseminating data on key non-communicable disease risk factors (e.g., tobacco use, alcohol use, physical inactivity, unhealthy diet, etc.) [21].

The level of commitment to a Healthy Lifestyle (HLS) was assessed on the basis of WHO recommendations [22,23]. A high commitment was defined as having a low level of physical activity, salt-free diet, or a low level of salt intake; sufficient consumption of vegetables and fruits (≥ 400 g/day); absence of smoking; and limited alcohol consumption (less than several times per week with a dose of ≤ 168 g per drink). A satisfactory level of commitment to an HLS was determined as the absence of smoking plus one additional HLS component listed above. A low level of commitment to an HLS was characterized by smoking. Those who did not comply with two or more of the HLS components in addition to smoking were also classified as having a low level of commitment.

In addition, participants were surveyed regarding the presence of workplace exposure risk factors: noise, vibration, unpleasant smells, and/or uncomfortable temperature (summer overheating, winter cooling). The workers were asked whether there had been any unscheduled visits to a doctor in the past year, whether they had had to take a temporary disability certificate, or whether they had been hospitalized. The total burden of the risk factors was examined by the railway for which the surveyed person worked. Statistical analysis of the results was carried out using the Kruskal–Wallis method and the chi-square criterion using the Excel 2019 and Statistica 13.0 programs.

## 3. Results

Significant differences were found in the frequency of workplace exposure risk factors between the railways of the surveyed participants (Table 1). The lowest frequency of workplace exposures was found in workers of the Trans-Baikal railway (railway 4), and the highest frequency was found for workers of the Krasnoyarsk, North, and South-East railways (railways 6, 12, 14).

The participants from the Far East and October railways (railways 3, 9) reported the lowest frequency of behavioral risk factors (Table 2). This frequency was highest for the participants from the Trans-Baikal railway (railway 4). It should be noted that from the risk factors analyzed, reliable differences between railways were found only for the frequency of alcohol abuse (*p* < 0.05), low consumption of vegetables and fruits (*p* < 0.05), and low levels of physical activity (*p* < 0.05).

The participants from the Eastern Siberian (railway 1), October (railway 9), and Southern Urals (railway 15) railways had the lowest frequency of measured individual health conditions (see Table 3), and the highest frequency was among participants from Krasnoyarsk (railway 6). Statistically significant differences were found for all of the parameters analyzed (*p* < 0.05).

The participants from the East Siberian (railway 1), Far East (railway 3), Kuibyshev (railway 7), and Sverdlovsk (railway 11) railways were the least likely to visit their doctor in the past year, need temporary disability, or be hospitalized (*p* < 0.05, see Table 4). The Northern railway (railway 12) had the highest combined rates for these medical activities (*p* < 0.05).

The total assessed HLS commitment index was higher for the participants from the Far Eastern (railway 3) and Southern Urals (railway 15) railways (*p* < 0.05, Table 5). Participants from the Moscow (railway 8) and October railways (railway 9) were the least committed to an HLS (*p* < 0.05). The total number of people with a high level of commitment to an HLS was 812, or 8.7% of the sample.

## 4. Discussion

In March 2019, the RF Government approved the long-term development program of the joint-stock company (JSC) “RZD” Russian Railways through to 2025, No. 466-r. It defines the planned values of key performance indicators of the JSC “RZD”. One of the key indicators of the program is maintaining the dynamics of the average annual rate of productivity growth of JSC “RZD” from 2019 to 2025 at the level of 105%. The list of enterprise-wide solutions contributing to the achievement of key indicators includes ensuring social stability and minimizing human resource risks.

From this perspective, the workforce is regarded as a critical asset [24]. Taking into account the scope of activity and status of JSC “RZD” as the largest employer in the country, the main priorities are the implementation of balanced personnel and social policy by providing a modern social package, as well as health and health promotion services [25]. Based on this, the realization of social policy and the maintenance of healthy staff at a level corresponding to the requirements of traffic safety is a strategic task of the social and personnel policy of the JSC “RZD” [26].

It should be noted, however, that in addition to improving the quality of care provided and increasing investment in this area, the annual average frequency and duration of temporary disability due to illness remains unchanged (49.2, 48.8, and 49.2 days in 2016, 2017 and 2018, respectively, per 100 employees) [26]. This may be due to the introduction of effective, constantly updated diagnostic methods in the JSC “RZD”, leading to an increase in the early detection of diseases. Members of train crews are a decreed contingent that complies with the Order of the Ministry of Transport of the Russian Federation dated 19 October 2020, No.428, “On approval of the Procedure for conducting mandatory preliminary (upon admission to work) and periodic (during employment) medical examinations on railway transport”(earlier-Order of the Ministry of Railways of the Russian Federation of 29 March 1999 N 6Ts “On approval of the Regulation on the procedure for conducting mandatory preliminary, upon admission to work, and periodic medical examinations on federal railway transport”). At least once every 2 years, drivers and their assistants undergo assessments by a commissioned medical expert, on the basis of which the decision of whether an employee of the train crew is allowed or not allowed to work on the train is made.

Train drivers appear to be a unique working group [27]. The prevalence of non-communicable diseases train crew in the RF [28] and other countries have been reported previously [29]. High levels of non-communicable diseases were reported for the drivers working on other modes of transport: metro [30], bus [31], and truck [32]. Drivers have additional risk factors such as shift work in combination with possible overworking time [33]. For train drivers, noise can be discussed as an additional risk factor. The role of noise exposure in the development of ischemic heart diseases, stroke, and arterial hypertension has been previously demonstrated [34]. A systematic review demonstrated the effect of exposure to occupational noise on cardiovascular diseases [35].

Worldwide experience has shown that corporate health programs are effective in influencing the way of life of employees, which makes it possible to reduce human risk factors leading to illness and chronic diseases and to increase the efficiency of work, including its economic composition [36]. Investigation of risk factors for non-communicable diseases is the first step towards the development of prevention programs [37], which may reduce the burden of non-communicable diseases [38]. Such programs are the most relevant for working-age people [39,40]. In Japan [41], for example, studies have shown that corporate physical activity and sports programs for employees can improve working ability and interpersonal relations among company personnel [42].

Research in the United States [23] shows that employers who introduce healthy workplace practices contribute to the overall health and well-being of their employees, increase productivity, and retain skilled workers [43,44]. This helps in reducing absenteeism and spending on health. Staff members who participate in workplace health programs reduce risks to their health [45,46].

According to the Library of Corporate Employee Health Promotion Programs developed by the RF Ministry of Health in August 2019 (https://minzdrav.gov.ru/poleznye-resursy/natsproektzdravoohranenie/zozh; accessed on 1 April 2022), Sberbank of Russia achieved a significant reduction in the number of days of temporary disability per employee during the two years of implementation of the wellness program. The prevention program also achieved an image-based goal, positively influencing the employee’s attitude towards the employer [47]. Introducing preventive measures of the JSC «Siberian Coal Energy Company» in 2018 marked a decrease in the frequency and duration of the following diseases: respiratory diseases (18%), digestive diseases (13%), and cardiovascular diseases (29%) [48]. Accordingly, there has been a decrease both in the total number of detected non-communicable diseases and in the number of workers who have first-time diagnoses of non-communicable disease [49].

It follows from the above that the achievement of the JSC “RZD” goal of increasing labor productivity is directly related to the improvement of workers’ health, which determines the relevance of the development of co-operative health programs and the concept of formation of an HLS among employees of JSC holding “RZD”. Our study results demonstrate a low level of commitment of locomotive crew members compared to the RF population [50]. For example, among RF working-age men, 18.3% have a low level of physical activity (in train drivers—59.3%), 34% report smoking (in train drivers—36.2%), 16.8% have a high level of HLS (in train drivers—8.7%). The low level of commitment to an HLS and the high prevalence of smoking in train drivers are particularly critical. Ultimately, the obtained results support efforts towards a more targeted development of corporate health programs for employees of the JSC “RZD”. Such programs in the holding company were introduced starting in 2020 and are in the process of being implemented.

This was the first study of risk factors and an HLS in train drivers in the RF. Of note, we were able to collect data from 7% of train drivers across the entire country. However, this study is limited by a single timepoint of investigation. We only studied train drivers from the JSC “RZD”, which is the biggest railway in the RF. We did not study train drivers from the other railway companies in the RF, although they only number about 10% of the total number of train drivers in the JSC “RZD”.

## 5. Conclusions

Locomotive crew members have a low level of commitment to the basic elements of an HLS. Occupational risk factors, behavior risk factors, and individual health conditions varied significantly by railway. The findings of the work warrant the establishment of personalized prevention programs for train drivers and their assistants.

## Figures and Tables

**Table 1 ijerph-19-07415-t001:** Workplace exposure risk factors by railway.

Railway	Noise n (%)	Vibration n (%)	Odor n (%)	Temperature n (%)
1	534 (60.27)	563 (63.54)	452 (51.02)	627 (70.77)
2	300 (55.45)	300 (55.45)	260 (48.06)	427 (78.93)
3	198 (62.26)	192 (60.38)	172 (54.09)	233 (73.27)
4	285 (42.54)	310 (46.27)	199 (29.70)	424 (63.28)
5	361 (51.42)	359 (51.14)	315 (44.87)	478 (68.09)
6	232 (63.56)	233 (63.84)	190 (52.05)	274 (75.07)
7	349 (60.17)	349 (60.17)	311 (53.62)	438 (75.52)
8	355 (50.35)	385 (54.61)	314 (44.54)	491 (69.65)
9	434 (51.61)	431 (51.25)	372 (44.23)	595 (70.75)
10	345 (60.63)	372 (65.38)	327 (57.47)	438 (76.98)
11	461 (56.08)	458 (55.72)	400 (48.66)	558 (67.88)
12	401 (70.35)	424 (74.39)	350 (61.40)	449 (78.77)
13	378 (49.67)	417 (54.80)	320 (42.05)	591 (77.66)
14	266 (61.72)	261 (60.56)	195 (45.24)	339 (78.65)
15	302 (55.21)	306 (55.94)	253 (46.25)	353 (64.53)

Notes: *p*-values were as follows, Noise: *p* < 0.001; Vibration: *p* < 0.001; Odor: *p* < 0.001; Temperature: *p* = 0.003; Abbreviations: n, number; %, percent.

**Table 2 ijerph-19-07415-t002:** Self-reported behavior risk factors for crew members by railway.

Railway	Smoking n (%)	Alcohol Intake n (%)	Low Fruit/Vegetable Intake (< 400 g/day) n (%)	High Processed Food Intake n (%)	High Intake of Ready-made Foods n (%)	Meals in Fast Food Restaurants n (%)	High Salt Intake (> 10 g/day) n (%)	Low Level of Physical Activity n (%)
1	303 (30.42)	565 (63.77)	359 (40.52)	809 (91.31)	778 (87.81)	531 (59.93)	416 (46.95)	499 (56.32)
2	196 (36.23)	323 (59.70)	154 (28.47)	498 (92.05)	454 (83.92)	278 (51.39)	252 (46.58)	312 (57.67)
3	110 (34.59)	173 (54.40)	105 (33.02)	275 (86.48)	266 (83.65)	183 (57.55)	149 (46.86)	210 (66.04)
4	269 (40.15)	387 (57.76)	283 (42.24)	597 (89.10)	569 (84.93)	349 (52.09)	276 (41.19)	408 (60.90)
5	227 (32.34)	427 (60.83)	187 (26.64)	652 (92.88)	574 (81.77)	337 (48.01)	351 (50.00)	390 (55.56)
6	133 (36.44)	254 (69.59)	104 (28.49)	331 (90.68)	305 (83.56)	168 (46.03)	164 (44.93)	186 (50.96)
7	204 (35.17)	371 (63.97)	157 (27.07)	526 (90.69)	497 (85.69)	287 (49.48)	250 (43.10)	349 (60.17)
8	282 (40.00)	424 (60.14)	220 (31.21)	657 (93.19)	570 (80.85)	390 (55.32)	353 (50.07)	400 (56.74)
9	313 (37.22)	542 (64.45)	287 (34.13)	790 (93.94)	701 (83.35)	495 (58.86)	374 (44.47)	430 (51.13)
10	210 (36.91)	318 (55.89)	119 (20.91)	509 (89.46)	483 (84.89)	290 (50.97)	219 (38.49)	358 (62.92)
11	326 (39.66)	530 (64.48)	267 (32.48)	757 (92.09)	700 (85.16)	430 (52.31)	332 (40.39)	518 (63.02)
12	210 (36.84)	405 (71.05)	169 (29.65)	513 (90.00)	483 (84.74)	301 (52.81)	247 (43.33)	373 (65.44)
13	289 (37.98)	431 (56.64)	169 (22.21)	693 (91.06)	592 (77.79)	350 (45.99)	343 (45.07)	481 (63.21)
14	140 (32.48)	244 (56.61)	115 (26.68)	396 (91.88)	339 (78.65)	180 (41.76)	196 (45.48)	245 (56.84)
15	178 (34.59)	323 (59.70)	296 (54.11)	493 (90.13)	466 (85.19)	250 (45.70)	211 (38.57)	337 (61.61)

Notes: *p*-values were as follows, Smoking: *p* = 0.09; Alcohol intake: *p* = 0.0036; Low fruit/vegetable intake: *p* < 0.001; High processed food intake: *p* = 0.103; High intake of ready-made foods: *p* = 0.066; Meals in fast food restaurants: *p* = 0.087; High salt intake: *p* = 0.078; Low level of physical activity: *p* = 0.015. Abbreviations: n, number; %, percent.

**Table 3 ijerph-19-07415-t003:** Measured individual health conditions by railway.

Railway	BMI > 25 kg/m^2^ n (%)	Waist Circumference > 94 cm n (%)	Blood Pressure > 130/80 mm/Hg n (%)	Resting Heart Rate > 80 Beats/min n (%)	Cholesterol > 5 mmol/L n (%)	Glucose > 5.5 mmol/L n (%)
1	496 (55.98)	117 (13.21)	48 (5.42)	99 (11.17)	91 (10.27)	40 (4.51)
2	318 (58.78)	146 (26.99)	69 (12.75)	104 (19.22)	70 (12.94)	24 (4.44)
3	188 (59.12)	44 (13.84)	44 (13.84)	74 (23.27)	67 (21.07)	23 (7.23)
4	381 (56.87)	117 (17.46)	72 (10.75)	80 (11.94)	114 (17.01)	61 (9.10)
5	457 (65.10)	216 (30.77)	79 (11.25)	105 (14.96)	140 (19.94)	46 (6.55)
6	247 (67.67)	98 (26.85)	58 (15.89)	71 (19.45)	60 (16.44)	30 (8.22)
7	371 (63.97)	122 (21.03)	69 (11.90)	75 (12.93)	33 (5.69)	10 (1.72)
8	463 (65.67)	114 (16.17)	116 (16.45)	102 (14.47)	93 (13.19)	37 (5.25)
9	466 (55.41)	156 (18.55)	83 (9.87)	92 (10.94)	146 (17.36)	57 (6.78)
10	364 (63.97)	155 (27.24)	56 (9.84)	95 (16.70)	93 (16.34)	12 (2.11)
11	490 (59.61)	173 (21.05)	73 (8.88)	158 (19.22)	79 (9.61)	42 (5.11)
12	372 (65.26)	119 (20.88)	65 (11.40)	65 (11.40)	111 (19.47)	78 (13.68)
13	455 (59.79)	156 (20.50)	99 (13.01)	152 (19.97)	228 (29.96)	80 (10.51)
14	281 (65.20)	126 (29.23)	62 (14.39)	74 (17.17)	59 (13.69)	18 (4.18)
15	333 (60.88)	61 (11.15)	52 (9.51)	89 (16.27)	39 (7.13)	6 (1.10)

Notes: *p*-values were as follows, BMI: *p* = 0.039; Waist Circumference: *p* < 0.001; Blood Pressure: *p* = 0.027; Resting Heart Rate: *p* < 0.001; Cholesterol: *p* < 0.001; Glucose: *p* < 0.001. Abbreviations: n, number; %, percent.

**Table 4 ijerph-19-07415-t004:** Self-reported medical activity during last year by railway.

Railway	Visited Their Doctor n (%)	Needed Temporary Disability n (%)	Was Hospitalized n (%)
1	295 (33.30)	258 (29.12)	73 (8.24)
2	201 (37.15)	181 (33.46)	38 (7.02)
3	105 (33.02)	100 (31.45)	27 (8.49)
4	244 (36.42)	217 (32.39)	72 (10.75)
5	238 (33.90)	226 (32.19)	80 (11.40)
6	186 (50.96)	182 (49.86)	43 (11.78)
7	214 (36.90)	197 (33.97)	37 (6.38)
8	308 (43.69)	319 (45.25)	84 (11.91)
9	381 (45.30)	378 (44.95)	93 (11.06)
10	236 (41.48)	185 (32.51)	65 (11.42)
11	296 (36.01)	280 (34.06)	52 (6.33)
12	289 (50.70)	287 (50.35)	61 (10.70)
13	344 (45.20)	299 (39.29)	86 (11.30)
14	164 (38.05)	172 (39.91)	52 (12.06)
15	203 (37.11)	183 (33.46)	44 (8.04)

Notes: *p*-values were as follows, Visited their doctor: *p* < 0.001; Needed temporary disability: *p* < 0.001; Was hospitalized: *p* = 0.027. Abbreviations: n, number; %, percent.

**Table 5 ijerph-19-07415-t005:** Level of commitment to a healthy lifestyle by railway.

Railway	Low n (%)	Moderate n (%)	High n (%)
1	303 (34.20)	506 (57.11)	77 (8.69)
2	196 (36.23)	302 (55.82)	43 (7.95)
3	110 (34.59)	165 (51.89)	43 (13.52)
4	269 (40.15)	328 (48.96)	73 (10.90)
5	227 (32.34)	425 (60.54)	50 (7.12)
6	133 (36.44)	198 (54.25)	34 (9.32)
7	204 (35.17)	322 (55.52)	54 (9.31)
8	282 (40.00)	375 (53.19)	48 (6.81)
9	313 (37.22)	477 (56.72)	51 (6.06)
10	210 (36.91)	299 (52.55)	60 (10.54)
11	326 (39.66)	431 (52.43)	65 (7.91)
12	210 (36.84)	303 (53.16)	57 (10.00)
13	289 (37.98)	404 (53.09)	68 (8.94)
14	140 (32.48)	256 (59.40)	35 (8.12)
15	178 (32.54)	315 (57.59)	54 (9.87)

Notes: *p*-values were as follows, Low: *p* = 0.094; Moderate: *p* = 0.018; High: *p* = 0.103. Abbreviations: n, number; %, percent.

## Data Availability

An Excel file is available from the first author upon request.

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
