# Peer review of "Risk Factors for Locomotive Crew Members Depending on Their Place of Work"

_ijerph, 2022, doi:10.3390/ijerph19127415_

Round 1

Reviewer 1 Report

Comments to authors

The paper has a well-established theoretical framework and well-organized content. However, the following modifications are recommended:

  1. In the introduction section, little explain has been made. There are at least a few research from the effects of various tasks on workers.
  2. The paper contains many examples of grammatical and spelling mistakes. Therefore, the entire manuscript must be revised to improve the language.
  3. Did the research undergo an ethics approval process? There is no mention of ethics approval number in the manuscript.
  4. In the discussion section, little comparison has been made between the findings of this study and those of the previous studies in the various workplace. A more thorough comparison is required.

Author Response

  1. In the introduction section, little explain has been made. There are at least a few research from the effects of various tasks on workers.

Corrected

  1. The paper contains many examples of grammatical and spelling mistakes. Therefore, the entire manuscript must be revised to improve the language.

The last author, Katie Heinrich is a native English speaker and has carefully proofread the manuscript.

English was corrected

  1. Did the research undergo an ethics approval process? There is no mention of ethics approval number in the manuscript.

In section “Material and methods” it is wrote “This study was approved by the inter-university ethics committee (Approval #07-19 from 18.07.2019).”

  1. In the discussion section, little comparison has been made between the findings of this study and those of the previous studies in the various workplace. A more thorough comparison is required.

Corrected

Reviewer 2 Report

According to the references, this group of authors (or a least a core group among them) has already published a lot on this issue at national level.

The effort to bring this study to an international level is certainly welcome, but it would need a little more work to make available to the international readers the background and the methodology used, as well as more structured and interesting analyses.

As an example, the authors declare that the methodology has already been described in detail in two previous studies but these are published in Russian language thus being accessible only to a Russian speaking audience, while the international readers may lack a lot of relevant information.

For this reason, the Materials and Methods section may benefit from a more detailed description of the research method.

In addition, there are no socio-demographic information available about the sample (gender, age, seniority of employment). Also, it is very difficult to understand what could be the differences in workplaces (the title states that risk factors are “depending on their places of work”): as the population is train drivers and train drivers assistants, one should think that the place of work is the locomotive (very similar if not identical across the entire sample). Thus, differences sit with the context of the different railways (e.g.: region, duration and length of routes, type of locomotive, climate along the route, shifts, work organizations, etc.) but no information (except for the name of the 15 railways) is disclosed throughout the paper.

The lack of these information makes this study very elementary and trivial and the analysis provided is just a simple reporting of frequencies of events or conditions with no insight on the effect of age, gender, working conditions, etc.

In my opinion, the study would be greatly improved by moving beyond this simple description of the data to include statistical modelling.

The results of the statistical analysis would also contribute to greatly improve the discussion and the conclusions of this study.

In the material and methods section, reference is made to the STEPS tool but searching the reference for this tool brings nowhere (it probably is a book chapter in Russian). If the tool used is the WHO STEPwise Approach to NCD Risk Factor Surveillance (STEPS), then the correct reference should be included (in addition to Boytsov)  

Some definitions need improvements: in table 1 the risk factor “sound” must be changed to “noise”

BMI, waist circumference, blood pressure, resting hearth rate, Cholesterol and Glucose should not be referred to as “biological risks”. In Occupational Safety and Health biological risks are those related to the exposure to biological agents (viruses, bacteria, fungi, parasites); “individual health conditions” may be an appropriate definition or anything you consider suitable other than biological risk factor.   

Author Response

For this reason, the Materials and Methods section may benefit from a more detailed description of the research method.

We added additional information in Material and Methods section

In addition, there are no socio-demographic information available about the sample (gender, age, seniority of employment). Also, it is very difficult to understand what could be the differences in workplaces (the title states that risk factors are “depending on their places of work”): as the population is train drivers and train drivers assistants, one should think that the place of work is the locomotive (very similar if not identical across the entire sample). Thus, differences sit with the context of the different railways (e.g.: region, duration and length of routes, type of locomotive, climate along the route, shifts, work organizations, etc.) but no information (except for the name of the 15 railways) is disclosed throughout the paper.

We added additional information

The lack of these information makes this study very elementary and trivial and the analysis provided is just a simple reporting of frequencies of events or conditions with no insight on the effect of age, gender, working conditions, etc.

We added additional information

In my opinion, the study would be greatly improved by moving beyond this simple description of the data to include statistical modelling.

The results of the statistical analysis would also contribute to greatly improve the discussion and the conclusions of this study.

Results of statistical analysis are present in the paper

In the material and methods section, reference is made to the STEPS tool but searching the reference for this tool brings nowhere (it probably is a book chapter in Russian). If the tool used is the WHO STEPwise Approach to NCD Risk Factor Surveillance (STEPS), then the correct reference should be included (in addition to Boytsov)  

We added reference information for the international publication. But in our research we used a Russian questionnaire, for which translation was made by Boytsov.

Some definitions need improvements: in table 1 the risk factor “sound” must be changed to “noise”

Changed

BMI, waist circumference, blood pressure, resting hearth rate, Cholesterol and Glucose should not be referred to as “biological risks”. In Occupational Safety and Health biological risks are those related to the exposure to biological agents (viruses, bacteria, fungi, parasites); “individual health conditions” may be an appropriate definition or anything you consider suitable other than biological risk factor.   

We have made this change throughout the paper.

Round 2

Reviewer 2 Report

Overall the paper has been improved, particularly in the introduction and discussion sections, that are integrated with new concepts and more international and updated references.

As the authors mentioned the study of Texeira et al (ref #34) on the prevalence of exposure to occupational noise, I suggest to also include the systematic review again from Teixeira et al on the effect of exposure to occupational noise on cardiovascular diseases. The estimates made available through this second study are even more consistent with this article.

The analysis remains very elementary; just frequencies of conditions with no modelling by age or by health condition or other variables. HOwever, this is the declared aim of the study, so - with the additional information provided - it can be fine. 

English language needs more (minor) revision

Author Response

Dear Sirs,

Thank you for your comments.

Overall the paper has been improved, particularly in the introduction and discussion sections, that are integrated with new concepts and more international and updated references.

As the authors mentioned the study of Texeira et al (ref #34) on the prevalence of exposure to occupational noise, I suggest to also include the systematic review again from Teixeira et al on the effect of exposure to occupational noise on cardiovascular diseases. The estimates made available through this second study are even more consistent with this article.

The reference is added

The analysis remains very elementary; just frequencies of conditions with no modelling by age or by health condition or other variables. HOwever, this is the declared aim of the study, so - with the additional information provided - it can be fine.

English language needs more (minor) revision

The last author, Katie Heinrich is a native English speaker and has carefully proofread the manuscript.